# Interleukin-17 and Th17 Lymphocytes Directly Impair Motoneuron Survival of Wildtype and FUS-ALS Mutant Human iPSCs

**DOI:** 10.3390/ijms22158042

**Published:** 2021-07-27

**Authors:** Mengmeng Jin, Katja Akgün, Tjalf Ziemssen, Markus Kipp, Rene Günther, Andreas Hermann

**Affiliations:** 1Department of Neurology, Technische Universität Dresden, 01307 Dresden, Germany; Jin.Mengmeng@uniklinikum-dresden.de (M.J.); Katja.Akguen@uniklinikum-dresden.de (K.A.); Tjalf.Ziemssen@uniklinikum-dresden.de (T.Z.); Rene.Guenther@uniklinikum-dresden.de (R.G.); 2Center for Clinical Neuroscience, University Hospital Carl Gustav Carus, 01307 Dresden, Germany; 3Institute of Anatomy, University Medical Center Rostock, Gertrudenstrasse 9, 18057 Rostock, Germany; Markus.Kipp@med.uni-rostock.de; 4Center for Transdisciplinary Neurosciences, University Medical Center Rostock, 18057 Rostock, Germany; 5German Center for Neurodegenerative Diseases (DZNE), 01307 Dresden, Germany; 6Translational Neurodegeneration Section, “Albrecht-Kossel”, Department of Neurology, University Medical Center Rostock, 18057 Rostock, Germany; 7German Center for Neurodegenerative Diseases (DZNE) Rostock/Greifswald, 18147 Rostock, Germany

**Keywords:** amyotrophic lateral sclerosis, induced pluripotent stem cells, motor neuron, multiple sclerosis, neurodegeneration, fused in sarcoma, Th17 cell, interleukin-17

## Abstract

Amyotrophic lateral sclerosis (ALS) is a progressive disease leading to the degeneration of motor neurons (MNs). Neuroinflammation is involved in the pathogenesis of ALS; however, interactions of specific immune cell types and MNs are not well studied. We recently found a shift toward T helper (Th)1/Th17 cell-mediated, pro-inflammatory immune responses in the peripheral immune system of ALS patients, which positively correlated with disease severity and progression. Whether Th17 cells or their central mediator, Interleukin-17 (IL-17), directly affects human motor neuron survival is currently unknown. Here, we evaluated the contribution of Th17 cells and IL-17 on MN degeneration using the co-culture of iPSC-derived MNs of fused in sarcoma (FUS)-ALS patients and isogenic controls with Th17 lymphocytes derived from ALS patients, healthy controls, and multiple sclerosis (MS) patients (positive control). Only Th17 cells from MS patients induced severe MN degeneration in FUS-ALS as well as in wildtype MNs. Their main effector, IL-17A, yielded in a dose-dependent decline of the viability and neurite length of MNs. Surprisingly, IL-17F did not influence MNs. Importantly, neutralizing IL-17A and anti-IL-17 receptor A treatment reverted all effects of IL-17A. Our results offer compelling evidence that Th17 cells and IL-17A do directly contribute to MN degeneration.

## 1. Introduction

Amyotrophic lateral sclerosis (ALS) is a late-onset neurodegenerative disease caused by the progressive loss of motor neurons (MNs) in the brain and spinal cord, leading to paralysis and eventually death within 1–5 years [1]. Most ALS cases are sporadic (sALS), while approximately 5–10% are familial (fALS), and the most prevalent mutated genes are ‘chromosome 9 open reading frame 72′ (C9orf72), ‘superoxide dismutase 1′ (SOD1), ‘fused in sarcoma’ (FUS), or ‘Tar DNA-binding protein 43′ (TDP-43) [2,3,4]. FUS mutations are among the most frequent genes found in both sALS and fALS [5,6] and account for the highest proportion of all gene mutations in juvenile ALS patients [7]. More than 100 FUS mutations have been found until now [8]. The P525L mutation was reported often and might concomitantly exhibit other phenotypes aside from the typical ALS phenotype, such as developmental disorders and movement impairments [9,10]. P525L was reported to be specifically associated with juvenile ALS onset and an aggressive disease course [11,12,13].

Although the pathophysiological mechanisms of ALS are still not fully understood, inflammatory mechanisms that contribute to the degeneration of ALS are reported and mainly attributed to central nervous system (CNS) innate immune cells [14,15,16]. MN loss is accompanied by microglia and astrocyte activation [17,18]. However, the contribution of the adaptive immune system in ALS is rather unknown. Preliminary data suggest the presence of CD4+ and CD8+ T cells in the spinal cord and brain of ALS [19] and T cell subset alterations in the peripheral blood of ALS patients [20,21,22]. The presence of adaptive immune cells in the CNS of ALS patients is in line with the idea that a brain-intrinsic degenerative process can trigger the recruitment of peripheral immune cells [23,24,25,26,27]. Interestingly, recent studies provide evidence of the relevant role of T helper (Th)17 cells and their activation, both in animal models and patients with ALS [22,28]. Furthermore, levels of the typical cytokine produced by Th17 cells, namely Interleukin-17A (IL-17A), were enhanced in the cerebrospinal fluid (CSF) and serum of ALS patients, too [29,30]. We also previously showed a Th1/Th17 shift in ALS patients related to ALS disease severity [31]. However, whether these T cell subsets can also directly affect MNs is currently unknown.

Patient-derived induced pluripotent stem cells (iPSCs), which are capable of self-renewal and differentiation into different cell types, offer remarkable possibilities for modeling human disease [32,33]. The ability to generate iPSCs from ALS patients and differentiate them into MNs has been a useful tool to study human neurons in vitro [34,35,36]. Here, we took advantage of in vitro ALS patient-specific, iPSCs-derived MNs carrying FUS mutations to delineate the direct impact of human Th17 cells and its main cytokine, IL17, on human MNs. For this, we developed a co-culture system of hiPSCs-derived MNs and Th17 cells, derived from the peripheral blood of ALS, healthy donors, and treatment-naïve multiple sclerosis (MS) patients. Moreover, we explored the influence of Th17 cell-produced cytokines, IL-17A and IL-17F, on MNs, including interference with possible therapeutic targets, such as the blockage of IL-17A or IL-17RA. Through this, we could show the critical role of Th17 cells and IL17A on human motor neuron survival.

## 2. Results

### 2.1. Th17 Cells Induce Cell Death in hiPSC-Derived MNs from ALS Patients

Recent reports suggested a neuropathological role of Th17 cells in animal models and patients with ALS [22,28], and IL-17A was enhanced in CSF and the serum of ALS patients [29,30]. In order to address whether Th17 directly affects MNs in ALS, we established a human co-culture system of pure hiPSC-derived MN and peripheral blood mononuclear cells (PBMCs)-derived Th17 cells (Figure 1A). MNs were generated from the hiPSCs of FUS-ALS patients and isogenic controls. To study disease-specific effects, we compared ALS patients to healthy, age-matched controls and immune-mediated disease multiple sclerosis patients (as a positive control). For details of the study population, please refer to Table 1.

We intended to be in similar ranges as would be found in vivo. Since concentrations of Th17 cells are unknown in ALS brain tissue, we approximated the proper Th17 to MN cell ratio as follows: in ALS, the postmortem tissue cell ratio between T lymphocytes and MNs is reported to be ~ 1:1–1:5 [37]. The percentage in the peripheral blood of ALS patients is about 1–2% of lymphocytes [31]. Thus, we tested effects in the range of 1:10–1:200 (Appendix A) and finally chose a concentration of 5 × 10^3^ Th17 cells that were seeded at the bottom of a well that was covered by a slip. The coverslip carried 3 weeks of matured MNs, which were seeded in a concentration of 1 × 10^5^ before (ratio Th17 cells: MNs 1:20). After the initiation of the co-culture, cells were co-incubated for 24 h. The co-culture of MNs with Th17 cells resulted in significant neuronal cell loss in the MS group assessed by MAP-2 positive neurons (Figure 1B,C). In contrast, no difference was observed regarding cell loss when comparing ALS patients or healthy donors with the control condition (without Th17 cells) (Figure 1B,C). Of note, the wildtype and FUS-ALS mutant MNs showed similar results.

Since the process of MN degeneration follows a dying back mechanism from the synapse to the perikaryon [38], we next assessed the neurite length of human MNs. The most striking effects were once again seen after co-culture with Th17 cells in the MS group for both the wildtype and FUS-ALS mutant MNs (Figure 1D). Of note, the healthy donor- and ALS patient-derived Th17 cells induced a significantly reduced neurite length in wildtype MNs, while there was only a trend in the case of FUS-ALS MNs (Figure 1D). Nevertheless, all the results imply that Th17 cells do have a detrimental effect on MNs (for detailed results please refer to the Supplementary Materials Appendix A). Additionally, we analyzed cytokines from the supernatants of the co-cultures system and found increased levels of IL-17 in MS and ALS co-cultures (Appendix A).

### 2.2. IL-17A Induces Neurodegeneration in WT and FUS-ALS MNs

IL-17 is often referred to as IL-17A, is the most widely investigated subtype in different diseases, and is also a signature cytokine of Th17 cells [39]. To study whether IL17A is the cytokine mediating the effect on human MNs, we used different concentrations of IL-17A (5 ng/mL, 50 ng/mL, and 500 ng/mL) to treat the MNs for three days. In line with the results of the MN-Th17 cell co-culture, we observed that the cell viability of WT and FUS MNs was reduced with increasing concentrations of IL-17A (Figure 2A,B). Similarly, IL17A induced a dose-dependent neuronal cell loss (Figure 2C) and neurite degeneration (Figure 2D). Of note, wildtype and FUS-ALS MNs showed similar vulnerability against IL17A (Figure 2A–D).

To look for long-term effects, we used lower concentrations of IL-17A (5 ng/mL) to treat the MNs for three and six days, respectively, causing significant neuron loss after six days with no differences between WT and FUS-ALS MNs (Figure 2E–G).

### 2.3. No Induction of Neurodegeneration Following IL-17F Treatment in WT and FUS-ALS MNs

IL-17F belongs to the IL-17 family (IL-17A–F) and is the other important cytokine that can be secreted by Th17 cells [40]. IL-17F and IL-17A can bind to the same receptor (IL-17RA and IL-17RC). To evaluate whether IL-17F has similar effects on MN survival, we used the same concentrations (5 ng/mL, 50 ng/mL, and 500 ng/mL) of IL-17F to treat MNs for three days. Surprisingly, cell viability, MN survival, and neurite length remained unaffected following IL17F treatment (Figure 3B–D). Therefore, despite sharing great sequence homology, IL-17F appears not to affect human MNs directly.

### 2.4. MNs Express the IL-17 Receptor (IL-17RA and IL-17RC)

IL-17A and IL-17F have 50% homology; they bind to the same receptor, comprising IL-17RA and IL-17RC [41]. Of note, there is currently no clear evidence as to whether these receptors are expressed on human neurons in principal, and specifically on MNs. To evaluate the expression of the IL-17 receptors, we performed IL-17RA, IL-17RC, and major histocompatibility complex class I (MHCI) staining on MNs, using PBMCs as positive and fibroblast as negative controls, respectively (Figure 4A). By doing so, we clearly showed that human iPSC-derived MNs do express IL17RA, IL17RC, and MCHI (Figure 4A,B). There was no difference between IL-17RA and the IL-17RC expression on WT and FUS MNs, but the MHCI was reduced in FUS MNs compared to WT MNs (Figure 4D).

### 2.5. IL-17A Induced MN-Neurotoxicity Was Neutralized by Treatment with Anti-IL-17A/IL-17RA

To prove whether IL17A signaling was mediated by IL17RA, we pretreated MNs in the co-culture with an anti-IL-17A neutralizing antibody (0.5 µg/mL) or an anti-IL-17RA antibody (1 µg/mL) for 0.5 h before the 50 ng/mL IL-17A treatment was initiated. Both the neutralizing IL17 reagent and the anti-IL17A antibody themselves had no deleterious effect on motor neurons (Figure 5). Both substances completely rescue the IL17A-mediated neurotoxic effects on both the WT and FUS-ALS MNs.

## 3. Discussion

Increasing evidence indicates that neuroinflammation contributes to the pathogenesis of ALS and may precede MN cell death [42,43,44]. The activation of microglia and the infiltration of lymphocytes surrounding MNs highlight the significant role of neuroinflammation in MN degeneration. Recent studies demonstrated that Th17 cell numbers are increased in the peripheral blood of ALS patients, and IL17 levels are increased in the CSF and serum of ALS patients, too. However, whether these cells or cytokines can directly impair human MNs is not known. By using patient-specific, iPSC-derived MN cell cultures, devoid of glial and microglial cells and co-cultured with patient-derived Th17 cells, we: (i) provide clear evidence of a direct, deleterious effect of Th17 cells and IL17A on human MNs; (ii) show that MNs express both IL17RA and IL17 RC; and (iii) most importantly, that these effects could be rescued by IL17 neutralization or anti-IL17RA treatment, thus offering novel and exciting drug targets for (motor) neurodegenerative diseases.

Our data fit well with recent reports, which indicated that Th17 cells and Th17-produced or related cytokines are elevated in ALS patients [22,29,30]. Even though IL-17A has the most sequence homology with IL-17F, which is also produced by Th17 cells [40,45], IL17F had no direct effect on human MNs. This might be due to the ~ 1000-fold higher affinity of IL17A to IL-17RA [46]. Since we neither measured a relevant amount of IL17A in pure MN supernatants (Supplementary Materials Appendix A, control condition) nor did IL17 neutralization/anti-IL17RA treatment show effects on MN survival, we can exclude that the MN culture already produces relevant amounts of IL17A on its own. However, we cannot exclude that higher concentrations of IL17F might have shown effects similar to IL17A. Nor can we exclude that other IL17 family members do also exert direct effects on MNs, such as IL17C. However, while IL-17A and F are well-known effectors of CD4+ T cells, IL-17C is expressed in a broader kind of tissue (e.g., epithelial cells). Therefore, we conclude that IL-17A plays a critical role in the effect of Th17 on MNs

Not much is known about the role of IL17 on human neurons. Some recent reports have suggested a direct interaction with neurons forming antigen-independent immune synapse-like contacts with the adhesion molecule Lymphocyte function-associated antigen 1 (LFA1). These Th17 neuron cell contacts resulted in increased intracellular Ca2+ with subsequent Ca2+ overload and neuronal damage [47]. Additionally, a direct injury to neurons by Th17 cells through the Fas/ Fas Ligand (FasL) pathway was reported in a mouse Alzheimer’s disease model [48]. Here, however, we provide clear evidence for a direct effect on the MNs by the Th17 cells and/or IL17, since human MNs do express IL-17RA, IL-17RC, and MHCI (Figure 4), and IL17 effects could be restored through anti-IL17RA treatment (Figure 5). Recent research has indicated that the pathogenic role of Th17 cells in autoimmune diseases and therapeutic interventions, including targeting Th17 effector cytokines, has provided beneficial effects in regard to rheumatoid arthritis, Crohn’s disease, and MS [49,50,51]. IL-17 antagonists are an effective treatment for plaque psoriasis [52]. So far, there is no relevant data on targeting Th17 cells and/or IL17 in ALS or other conditions with (motor) neuron degeneration. This study affords unequivocal evidence and gives us a hint that therapeutic interventions on Th17 cells and/or IL17A may be a promising treatment strategy beyond motor neuron degenerative disorders.

Of note, while we showed clear evidence of MN damage by Th17 cells and their cytokine IL17A, MNs from a familial ALS form did not show higher vulnerability to IL17A than their isogenic wildtypes, nor did they show differences in IL-17RA and IL-17RC expression. Future studies should clarify whether this holds true only for this specific genetic ALS variant or whether this is a general finding in ALS models. Interestingly, however, is the fact that FUS-ALS MNs showed reduced expression of MHCI, even in our neuronal cultures that were devoid of astrocytes. This is of interest since MHCI was recently reported to protect MNs from astrocyte-induced cell death, and reduced MHCI expression makes these MNs susceptible to astrocyte-induced cell death [53].

Th17 and their cytokines might be implicated in other neurological disorders, as well. For example, elevated frequencies of IL-17-producing CD3+ CD8− cells have been demonstrated in Parkinson’s disease [54], and a polymorphism in the IL-17A gene was shown to increase the risk of cognitive impairment in PD [55]. Although CD8+ cytotoxic T cells outnumber CD4+ lymphocytes in MS lesions, CD4+ cells are also present in actively demyelinating MS lesions, primarily in perivascular spaces and the meninges. There is no doubt that these cells can initiate the inflammatory process, activate effector microglia and macrophages, and stimulate MHC upregulation in the CNS. Since effects on wildtype MNs are as large as on ALS MNs, and Th17 cells from MS patients exerted the strongest effect concerning neuron loss and neurodegeneration (Figure 1), our study also has a significant impact on the overt question of neurodegeneration/axon degeneration in multiple sclerosis. MS patients were significantly younger than ALS patients and healthy controls, who had been matched to ALS patients. Of note, there is evidence that, even though the frequency of Th17 cells is lower in elder patients, Th17 cells are more differentiated and produce more IL-17 in the elderly [56]. However, further studies are needed to reveal the relevance of the IL17 cytokine family for neurodegeneration in MS in more detail.

## 4. Materials and Methods

### 4.1. Patient Material

In this study, we included patient material for (i) peripheral immune cell derivation and (ii) human iPSCs.

For (i), peripheral blood was taken from 5 ALS patients, 4 control subjects, and 4 treatment-naïve MS patients. The use of blood derivatives was approved by the local ethics committee (EK393122012 – approval date 11.03.2013, EK49022016 – approval date 26.01.2016, EK348092014 – approval date 25.11.2014). We included patients with definite, probable, or possible ALS, according to the revised El Escorial criteria [49]. Patients suffering from genetically proven spinal muscular atrophy, spinal bulbar muscular atrophy, and FTD overlap syndromes were excluded. Details on patients and CSF characteristics are shown in Table 1.

For (ii), we included an isogenic CRISPR/Cas9-derived hIPSC pair of wildtype and FUS-P525L ALS. The lines were fully characterized as published previously [38].

### 4.2. Peripheral Blood Mononuclear Cells (PBMCs) Isolation

PBMCs were isolated from the peripheral blood of different donors using a LeucosepTM (Greiner Bio-One). The blood was put into the LeucosepTM with 15 mL Biocoll (Biochrom GmbH, Berlin, Germany) and was centrifuged at 1000 g for 10 min at room temperature (RT). PBMC pellets were washed once with phosphate-buffered saline (PBS), centrifuged at 250 g for 10 min at 4 °C, and then counted directly. After that, the PBMCs were seeded at the proper density and incubated in the incubator at 37 °C and 5% CO_2_ for further use.

### 4.3. Th17 Cell Enrichment

The sorting of PBMCs from the fresh peripheral blood of different donors started by seeding them at the density of 1 × 10^7^ in a 6-well plate and incubating them overnight. The next day, cells were stimulated with CytoStim (Miltenyi Biotec, Bergisch Gladbach, Germany) for 3 h [57], followed by the application of the IL-17 Secretion Assay-Cell Enrichment and Detection Kit (Miltenyi Biotec, Bergisch Gladbach, Germany)—according to the manufacturer’s instructions—including magnetic cell separation with the program “Posseld”.

### 4.4. Genotyping

All iPSC cells were already derived and fully characterized in previous studies [35,36,38], including iPSCs from FUS-ALS patients, which had been isogenically corrected using CRIPS/Cas9n.

### 4.5. Differentiation of Human Neural Progenitor Cells (NPCs) to MNs

The differentiation of neural progenitor cells (NPCs) to MNs was accomplished following the protocol as previously described [38]. NPCs were kept on Matrigel-coated plates, and it was necessary to split them at least once a week at a ratio of 1:10. For the initiation of differentiation, the cells were cultured in N2B27 (DMEM-F12/Neurobasal 50:50 with 1:200 N2 Supplement, 1:100 B27 lacking Vitamin A and 1% penicillin/streptomycin/glutamine) containing 150 µM ascorbic acid, 3 µM CHIR 99021, and 0.5  µM pumorphamine (PMA). The MNs’ differentiation was induced by the N2B27 consisting of 0.2 mM ascorbic acid, 1 µM Retinoic Acid, 1 ng/mL brain-derived neurotrophic factor (BDNF), 1 ng/mL glial cell-derived neurotrophic factor (GDNF), and 0.5 µM smoothened agonist (SAG), and the differentiation medium was changed every second day. On day 8, the cells were split again and then seeded at the desired density and kept in a maturation medium consisting of N2B27 composed of 5 ng/mL Activin A (only for the first day), 2 ng/mL BDNF, 0.1 mM cAMP, 0.2 mM ascorbic acid, 1 ng/mL transforming growth factor-β (TGF-β), and 2 ng/µL GDNF. After maturation, the MNs were used for further experimentation.

### 4.6. Co-Culture of MNs with Th17 Cells

During the maturation of MNs, the medium was changed every second day before treatment. To develop the co-culture experiment, Th17 cells were sorted according to the manufacturer’s instructions (see above). In the co-culture systems, the MNs grown in Matrigel pre-coated slides of 24-well plates with MNs maturation medium and Th17 cells kept in a culture medium consisting of complete RPMI-1640 (Biochrom) without IL-2, with 5% human serum (CC pro), 100 U/mL penicillin, and 100 µg/mL streptomycin (Biochrom). Following sorting, Th17 cells were centrifuged at 500 g for 5 min, resuspended in complete RPMI-1640 media at a density of 10 µL/10^3^, and were put in a new 24-well plate. Slides with MNs were then transferred to the same wells containing 30 µL of Th17 cell suspension. Finally, 500 µL of fresh maturation MN media was added to each well, and the cells were co-incubated for 24 h. Subsequently, MNs were used for downstream experiments, and supernatants from the co-cultures were collected for ELISA.

### 4.7. Treatment of MN with IL-17 Subtypes

The MNs were stimulated with different concentrations of IL-17A (Adipogen, Hamburg, Germany) or IL-17F (R&D Systems, Minneapolis, MN, USA) and cultured for three days. Further, 0.5 µg/mL of IL-17A neutralizing antibody (1:1,000, BPS Bioscience, San Diego, CA, USA) and 1µg/mL of human IL17RA/IL-17R antibody pretreatment (1:500, R&D Systems, Minneapolis, MN, USA) for 0.5 h prior to adding IL-17A was performed for three days. After different treatments, the MNs were used for further immunostaining and the cell viability tests.

### 4.8. Immunofluorescence Staining and Neurite Network Analysis

Cells were fixed in 4% paraformaldehyde (PFA) in PBS for 15 min at room temperature (RT), followed by blocking with 5% donkey serum in PBS with 0.02% Triton-X100 for 1 h at RT. Then, the cells were incubated with rabbit anti-MAP2 (1:500, Millipore, Billerica, MA, USA), mouse anti-MAP2 (1:500, BD Pharmingen, San Diego, CA, USA), mouse anti-IL-17RA (1:1000, R&D Systems, Minneapolis, MN, USA), mouse anti-IL-17RC (1:500, Abcam, Cambridge, UK), rabbit MHCI (1:500, Abcam), rabbit anti-cleaved caspase 3 (1:500, Cell Signaling Technology), and chicken anti-SMI32 (1:10,000, Covance) at 4°C overnight. Cells were thereafter washed with PBS three times and incubated with Alexa Fluor^®^ 555 or 647 donkey anti-rabbit IgG or donkey anti-mouse IgG or Alexa Fluor^®^ 647 (1:500, Abcam) in the dark for 1 h. Finally, the nuclei were stained with Hoechst (Invitrogen). Images were taken using a fluorescent microscope (Zeiss, Goettingen, Germany). In Brief, neurites were recognized by MAP-2 staining; after imaging the neuronal network, the Neuron J plugin (Version 1.52 p) in Image J (ImageJ-win64, open-source) was opened, the image was selected, then we converted the picture into an 8-bit image (Appendix A). Following this, we labeled all the neurites using the “adding trace” button. After the tracing was completed (Appendix A), a text file including the neurite length measurement data was generated. We normalized the data within each experiment to the control condition. For each analysis, we took at least 15 pictures at random spots for each condition of every experiment. More than 50 neurons were analyzed in each group, and all experiments were repeated at least three times (see also the Supplementary Materials Appendix A).

### 4.9. Cell Viability Assay

Cell viability was measured using the cell counting kit-8 (CCK-8) (Sigma-Aldrich, St. Louis, MO, USA). In brief, the differentiated MNs were seeded into a 24-well microplate. After the maturation of the MNs, they were incubated with different treatments. Subsequently, 50 µL of CCK-8 solution was added to 500 µL media per well and incubated at 37 °C for 4 h. The colorimetric assay was measured using a Tecan microplate reader (Crailsheim, Germany).

### 4.10. Cytokine Analysis

The supernatants were collected from the co-culture system for cytokine measurements. The cytokine concentration was evaluated using commercially available IL-17A ELISA kits (eBioscience) that were utilized according to the manufacturer’s instructions.

### 4.11. Statistical Analysis

All results are presented as mean ± standard deviation (SD). Statistical significance was analyzed using the paired t-test for two-group comparison or a one-way analysis of variance (ANOVA) followed by a Tukey post hoc analysis for multiple groups using GraphPad Prism (GraphPad Software Version 6.0). The significance level was set at *p* < 0.05.

## 5. Conclusions

In summary, we propose that Th17 cells and their main effector, cytokine IL17A, can directly induce neurodegeneration in human neurons and that this effect is most likely driven via IL-17RA. We also prove the therapeutic potential of anti-IL17 treatment to avoid these direct neurodegenerative effects. This has a significant impact not only on motor neuron diseases, in which IL17 was reported to be increased in serum and CSF, but also in the primary neurodegeneration in MS.

## Figures and Tables

**Figure 1 ijms-22-08042-f001:**
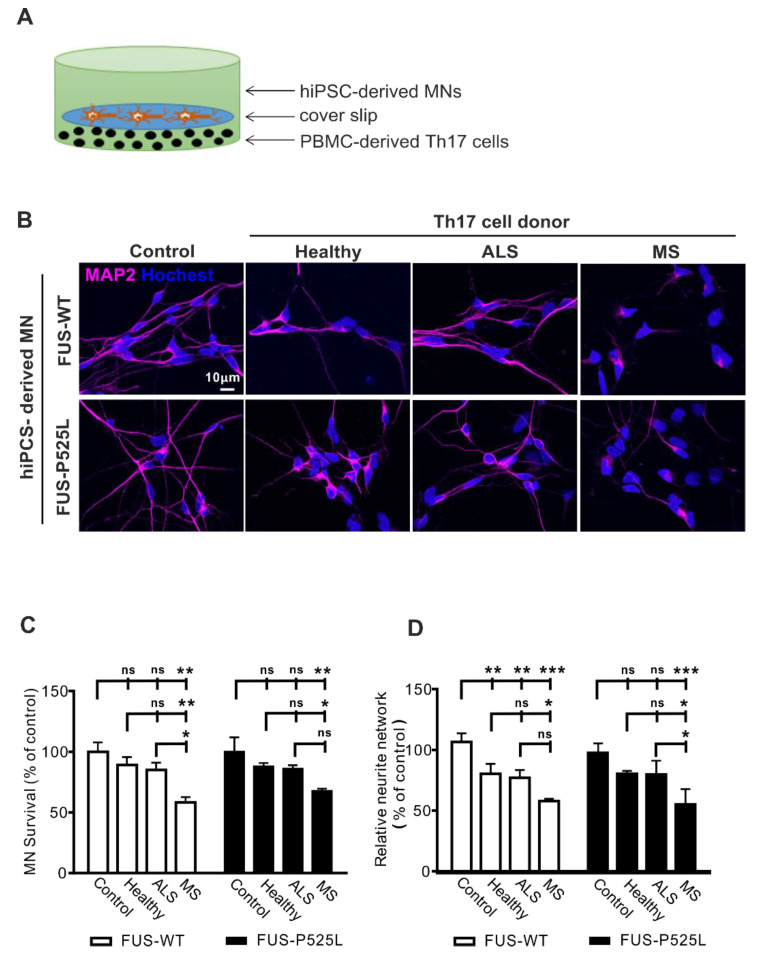
Increased cell death in ALS hiPSC-derived MNs after co-culture with Th17 cells from different donors (healthy donors, MS, and ALS patients) on MNs: (**A**) the scheme of Th17 cells co-culture with MNs; (**B**) immunostaining MNs with a MAP-2 antibody in co-culture with Th17 cells from healthy donors, ALS, and MS patients for 24 h; (**C**) the viability of MNs after different treatments was measured through CCK-8 assay; (**D**) the length of neurites were evaluated under different co-culture conditions using MAP-2 immunostaining, and the neurite networks were analyzed with Image J. * *p* < 0.05, ** *p* < 0.01, *** *p* < 0.001 versus controls, ns: no significance. For details, please refer to the Supplementary Materials Appendix A.

**Figure 2 ijms-22-08042-f002:**
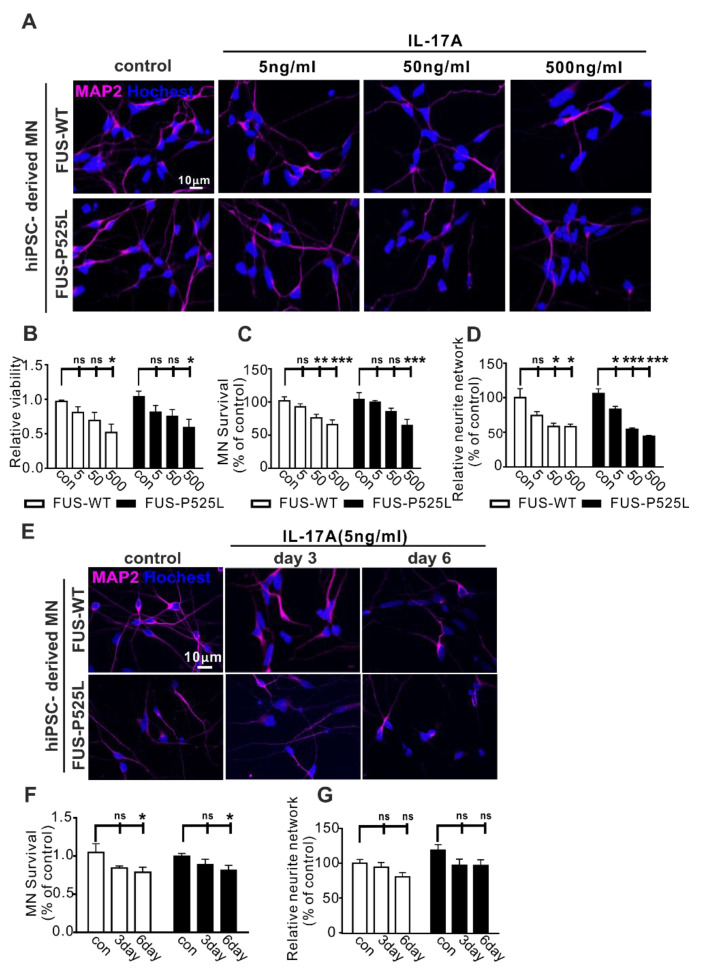
Impact of IL-17A on MNs: (**A**) the MAP-2 immunostaining after different concentrations (5 ng/mL, 50 ng/mL, and 500 ng/mL) of IL-17A-stimulated MNs for three days; (**B**) the viability of MNs under the same IL-17A treatment for three days was evaluated using CCK-8 assay; (**C**) the MAP-2 positive MNs, counting manually in different IL-17A treatments; (**D**) the neurite length of MAP-2 positive MNs was calculated using Neuron J in the FIJI software; (**E**) the immunostaining of MAP-2 with the same IL-17A treatment at different times (three days and six days); (**F**) the MAP-2 positive MNs’ counting in IL-17A treatment at different times; (**G**) the length of the neurite was measured under the same treatment. * *p* < 0.05, ** *p* < 0.01, *** *p* < 0.001 versus controls, ns: no significance.

**Figure 3 ijms-22-08042-f003:**
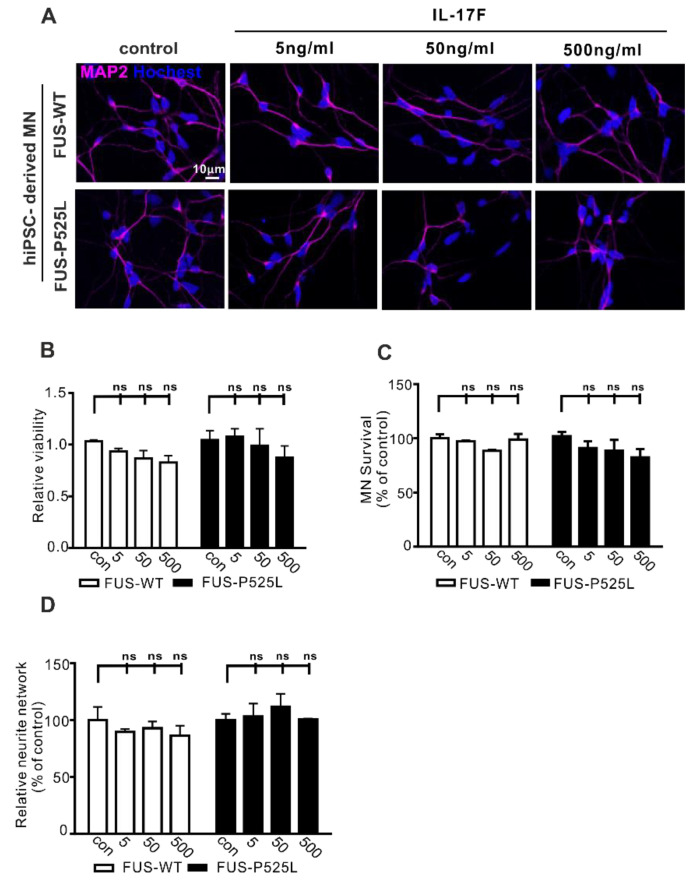
Effect of IL-17F on MNs: (**A**) immunostaining of MAP-2 in IL-17F (5 ng/mL, 50 ng/mL, and 500 ng/mL) stimulated MNs; (**B**) the viability of MNs under different IL-17F treatments was measured using CCK-8 assay; (**C**) MAP-2 positive MNs counting under different IL-17F treatments; (D) the neurite length was calculated, using Neuron J in the FIJI software, under different treatment conditions in both cells, ns: no significance.

**Figure 4 ijms-22-08042-f004:**
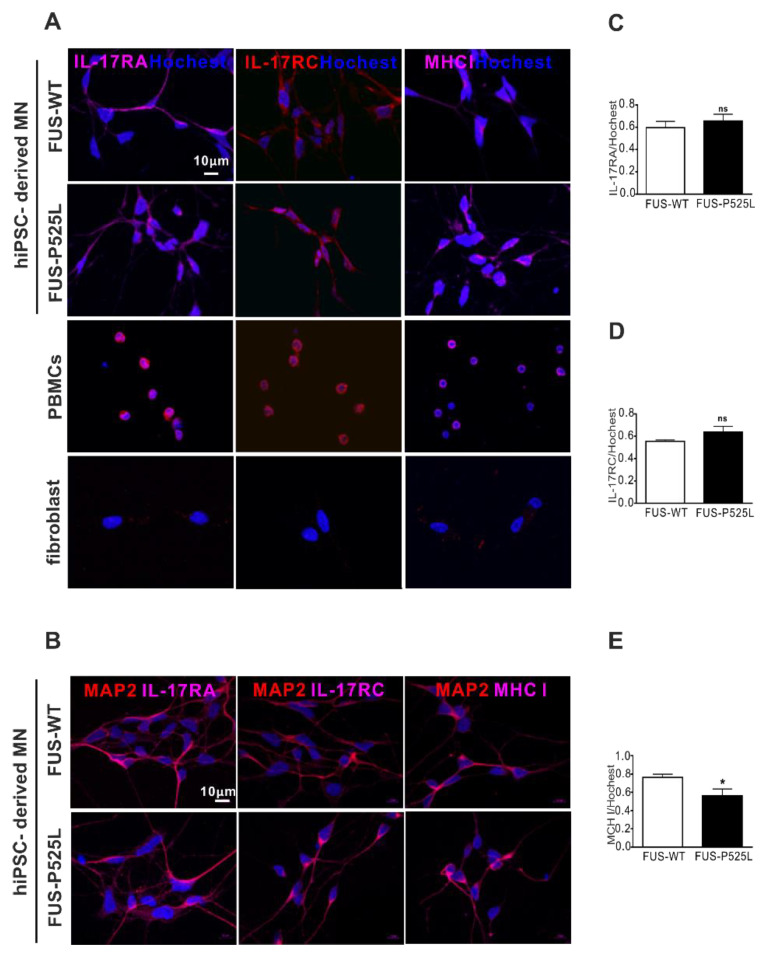
Immunostaining of the IL-17 receptor (IL-17RA and IL-17RC) and MHCI in MNs: (**A**) immunostaining of IL-17RA, IL-17RC, and MHCI in FUS-WT-EGFP, FUS-P525L-EGFP, PBMCs, and a fibroblast: (**B**) co-staining of MAP-2 with IL-17RA, IL-17RC, and MHCI in FUS-WT-EGFP and FUS-P525L-EGFP cells; (**C**) the counting of IL-17RA positive cells in MNs; (**D**) IL-17RA positive staining in MNs; (**E**) MHCI positive staining in MNs. * *p* < 0.05 versus controls, ns: no significance.

**Figure 5 ijms-22-08042-f005:**
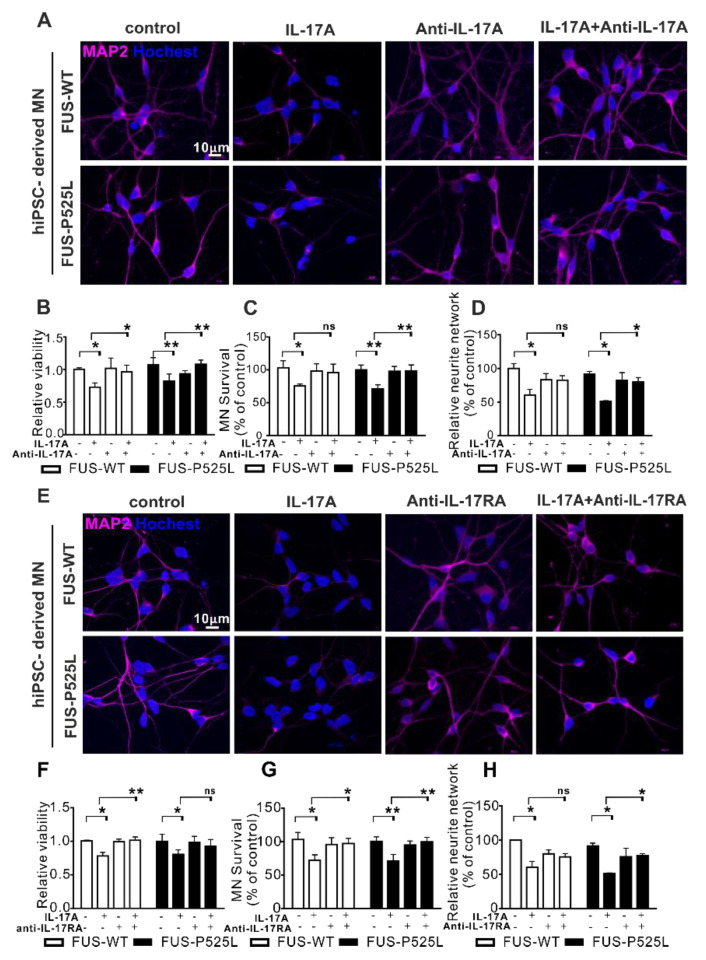
Antagonizing IL17 protects from IL17-induced neurodegeneration: (**A**) IL-17A-neutralizing pretreatment for 0.5 h, 50 ng/mL IL-17A stimulated the MNs for three days, MAP-2 immunostaining was performed after the treatment; (**B**) the viability of MNs under the same treatment mentioned before was calculated using CCK-8 assay; (**C**) MAP-2 positive MNs were counted and **(D)** the neurite length of MNs were measuredunder different treatments; (**E**) immunostaining of MAP-2 under anti-IL-17RA/IL-17R receptor pretreatment for 0.5 h, then 50 ng/mL IL-17A-stimulated for three days; (**F**) viability of MNs; (**G**) MAP-2 positive MNs counting; (H) neurite length of MAP-2 positive MNs using the aforementioned treatment.* *p* < 0.05, ** *p* < 0.01 versus controls, ns: no significance.

**Table 1 ijms-22-08042-t001:** Laboratory values of the used PBMCs and patients’ characteristics; data are depicted in mean (SD).

Patient Parameter	ALS	MS	Healthy Donors
Number	5	7	4
Gender m:f	2:3	1:6	3:1
Age	66.3 ± 11	27 ± 6	52.5 ± 19.8
ALS-specific			
ALS disease onset, spinal: bulbar	2:3	n.a.	n.a.
ALS sporadic, genetic, familiar	100/0/0%	n.a.	n.a.
ALSFRS-R (Range)	32.6 ± 10.8	n.a.	n.a.
MS subtype,%			
Relapsing-remitting	n.a.	100%	n.a.
Primary progressive	n.a.	0%	n.a.
Secondary progressive	n.a.	0%	n.a.

n.a.: not applicable.

## Data Availability

All data is presented in the article.

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
