# Peer review of "Interleukin-17 and Th17 Lymphocytes Directly Impair Motoneuron Survival of Wildtype and FUS-ALS Mutant Human iPSCs"

_ijms, 2021, doi:10.3390/ijms22158042_

Round 1
Reviewer 1 Report
Jin and colleagues present an interesting report describing the IL-17 and Th17 lymphocytes effects on the survival of motor neurons derived from iPSC from patients with ALS, MS, and healthy controls. The manuscript is well written, the study well presented, and the discussed topic relevant to improving our understanding of ALS pathology. My comments are listed below in the same order they occur throughout reading the manuscript.
- I assume that since IL-17A is the most abundant cytokine and IL-17F having the highest homology to IL-17A, the authors focused on these 2 cytokines. However, studying the other interleukins from the IL-17 family might have yielded interesting results, e.g., IL-17C was shown to have strong pro-inflammatory functions. Why were not these assessed?
- The authors focused on iPSCs that carry a specific mutation. However, it will also be interesting to know whether mutations in other genes linked to ALS will have a similar response to the effects of IL-17. Has this been investigated?
- It is unclear what additional information comparing the ALS to MS cells brings to the table. From the presented results, it seems that, if anything, the MS cells are the most affected cells by treatment with IL-17A. Based on these observations, one can argue that IL-17A has a generalized effect in a broader group of neurodegenerative disorders, but so what?
- It is unclear how and what source was the ALS iPSCs derived from. Furthermore, it is unclear why the iPSCs were not derived from the same patients from whom Th17 lymphocytes were extracted. The data interpretation and conclusions from the results would be more robust if both the iPSCs and lymphocytes originating from the same subjects, e.g., eliminating the confounding effect of different genetic backgrounds.
- Table 1 is confusing. I'm unsure what does n.d. for gender under the MS columns means. How are the authors unaware of the sex of MS subjects!?
- The rationale for choosing the seeding concentration for the different cells needs to be explained. If these reflect growth rate, i.e., a need for a certain level of cell confluency, that needs to be explained.
- 1D doesn't allow for any quantitative (or qualitative) assessment. Additional information is need about the algorithm parameters that were used to determine the neurite length. It will be helpful, if the mean with (±SD) of the number of neurites (from how many cells) per group, per visual field, are presented in table format.
Were the Ab used in the immunohistochemistry experiments mono- or poly-clonal? How was the labelling specificity assured?
Author Response
Reviewer 1
Comments and Suggestions for Authors
Jin and colleagues present an interesting report describing the IL-17 and Th17 lymphocytes effects on the survival of motor neurons derived from iPSC from patients with ALS, MS, and healthy controls. The manuscript is well written, the study well presented, and the discussed topic relevant to improving our understanding of ALS pathology.
Response: We appreciate this overall positive review.
My comments are listed below in the same order they occur throughout reading the manuscript.
- I assume that since IL-17A is the most abundant cytokine and IL-17F having the highest homology to IL-17A, the authors focused on these 2 cytokines. However, studying the other interleukins from the IL-17 family might have yielded interesting results, e.g., IL-17C was shown to have strong pro-inflammatory functions. Why were not these assessed?
Response: We appreciate this comment. As the reviewer stated we focused in this pilot study on the effect of Th17 cells and their main effectors IL-17A&F. We focused on the latter because of their homology, their well known expression by CD4+ T cells and their well known role in inflammatory settings. Instead, IL-17C is expressed of a broader kind of tissue (e.g. epithelial cells) and the definite source is not concluding clear. To avoid more speculative conclusions, we did not focus on IL-17C in this pilot study. However, this might be of interest in follow-up projects. We discussed this in the revised version of the discussion section.
- The authors focused on iPSCs that carry a specific mutation. However, it will also be interesting to know whether mutations in other genes linked to ALS will have a similar response to the effects of IL-17. Has this been investigated?
Response: We do for sure agree, however as we already stated in our manuscript, this should be the focus on follow up studies (please refer to lines 250ff of Discussion section).
- It is unclear what additional information comparing the ALS to MS cells brings to the table. From the presented results, it seems that, if anything, the MS cells are the most affected cells by treatment with IL-17A. Based on these observations, one can argue that IL-17A has a generalized effect in a broader group of neurodegenerative disorders, but so what?
Response: We do see the reviewer’s point. The intention (and power) for including the MS sample was more to have a clear positive control rather than specifically work on MS. However, since the effects shown in the manuscript are not influenced by ALS mutation but obvious both in ALS and even more MS samples, we suggest that Th17 cells and Il17A might be very exciting effectors of (motor) neurodegeneration with therapies being in principle already available for repurposing strategies.
Our intention with the table was to clearly depict the clinical cohort for the interested reader and thus kindly like to suggest to keep it in the manuscript.
- It is unclear how and what source was the ALS iPSCs derived from. Furthermore, it is unclear why the iPSCs were not derived from the same patients from whom Th17 lymphocytes were extracted. The data interpretation and conclusions from the results would be more robust if both the iPSCs and lymphocytes originating from the same subjects, e.g., eliminating the confounding effect of different genetic backgrounds.
Response: We do appreciate the comment. However, it is important to note, that we used CRISPR/Cas9 engineered isogenic iPSCs – the gold standard in iPSC research – but primary lymphocytes from ALS/MS patients and age-/sex matched controls. Even the suggestion of the reviewer is a very interesting approach, there are different stumbling blocks which need to be considered. A main difficulty is the timing, because the complete process of generating stable iPSCs from skin biopsy and stable differentiation/maturation to spinal motoneurons can take up to a half year. In contrast, immune cells need to be processed immediately and kept in culture. In view of the rapidly progressing disease implementation and timing both steps would be very challenging. Furthermore, we did not intend to work with iPSC-generated lymphocytes, which is at least in our view a completely different approach and storyline, since lymphocyte subsets are (1) hard to differentiate from iPSCs and (2) most importantly, most likely are not fully functional and matured as they are in vivo. Thus, even in the case we would have used lymphocytes from the parental ALS patients, we would have to use lymphocytes from age-/sex matched controls, which makes comparison as difficult.
- Table 1 is confusing. I'm unsure what does n.d. for gender under the MS columns means. How are the authors unaware of the sex of MS subjects!?
Response: Thanks a lot for the comment. We adapted the table concerning the missing data.
- The rationale for choosing the seeding concentration for the different cells needs to be explained. If these reflect growth rate, i.e., a need for a certain level of cell confluency, that needs to be explained.
Response: Thanks for this comment. Concerning MNs we used our standard differentiation protocols with defined seeding densities since these are important for proper differentiation and survival of MNs, in our case 1x105.
Concerning cell density of Th17 cells: We intended to be in similar ranges as it would be in vivo. First, the calculated cell ratio in ALS postmortem tissue between T lymphocytes and MNs is around 1:1-1:5 (1). No publication so far directly indicates how many Th17 cells are in ALS postmortem tissue. The percentage should be obviously much lower than T lymphocytes and MNs ratio in ALS brain tissue. Secondly, to get enough Th17 cells, we used CytoStim to develop rapid and efficient restimulation of human T cells. After the stimulation the CD4+ and CD8+ cells start to secrete effector cytokines within a few hours. We next sorted the Th17 cells using the IL-17 Secretion Assay-Detection Kit. In the end, we got the higher IL-17 secreting Th17 cells compared to unstimulated cells, therefore, based on these reasons, we test the ratio of Th17 cells and MNs at the range of 1:10-1:200, the results are shown in Fig. S1, according to the concentration test, we chosen the 5x103 Th17 cells together with 1x105 MNs. We added this in the results part of the revised manuscript.
- 1D doesn't allow for any quantitative (or qualitative) assessment. Additional information is need about the algorithm parameters that were used to determine the neurite length. It will be helpful, if the mean with (±SD) of the number of neurites (from how many cells) per group, per visual field, are presented in table format.
Response: Dear Reviewer, thank you very much for this comment. We added more detailed informations in the methods part and a supplement figure with an example of the NeuronJ analysis. Furtehrmore, we exemplarily add the detailed datasets of the analysis of Figure 1D in a suppl. Table.
- Were the Ab used in the immunohistochemistry experiments mono- or poly-clonal? How was the labelling specificity assured?
Response: The Abs used in the study to label neurons or motor neurons are well established from many previous publications of our group, the specificity of any doubt. Every experiment always included secondary AB only specimens to prove unspecific secondary stainings.
The stainings with the IL17 R and MHC were novel for us and thus included more controls. All these Abs were monoclonal. For positive and negative controls we included PBMC (known to express these receptors) and fibroblasts (known not to express these receptors), respectively (Figure 4A).
Reviewer 2 Report
The paper titled "Interleukin-17 and Th17 lymphocytes directly impair motoneuron survival of wildtype and FUS-ALS mutant human iPSCs" written by Jin et al. describes the IL-17 and Th17 lymphocytes effects on the survival of motor neurons derived from iPSC from patients with ALS, MS, and healthy controls.
Minor revisions:
- The authors should improve Introduction better clarifying the aim of the study.
- - Please include latest citations about paper topic
Author Response
Reviewer 2
Comments and Suggestions for Authors
The paper titled "Interleukin-17 and Th17 lymphocytes directly impair motoneuron survival of wildtype and FUS-ALS mutant human iPSCs" written by Jin et al. describes the IL-17 and Th17 lymphocytes effects on the survival of motor neurons derived from iPSC from patients with ALS, MS, and healthy controls.
Response: We deeply thank the reviewer for the very positive evaluation of our study.
Minor revisions:
- The authors should improve Introduction better clarifying the aim of the study.
Response: We did rewrite the introduction to better clarify the aim of the study
- Please include latest citations about paper topic
Response: We did thouroughly update our citaion list and add a couple of new papers in the revised manuscript.
Reference Point-by point:
- Kawamata T, Akiyama H, Yamada T, McGeer PL. Immunologic reactions in amyotrophic lateral sclerosis brain and spinal cord tissue. The American journal of pathology. 1992;140(3):691-707.
Round 2
Reviewer 1 Report
I have no further comments for the authors.